# Application of Amplicon-Based Targeted NGS Technology for Diagnosis of Drug-Resistant Tuberculosis Using FFPE Specimens

Jing Song,[a] Weili Du,[a] Zichen Liu,[a] Jialu Che,[a] Kun Li,[a] Nanying Che[a]

[a]Department of Pathology, Beijing Key Laboratory for Drug Resistant Tuberculosis Research, Beijing Chest Hospital, Capital Medical University, Beijing Tuberculosis and Thoracic Tumor Research Institute, Beijing, China

Jing Song, Weili Du, and Zichen Liu contributed equally to this article. Authors order is determined according to their contributions to the study.

**ABSTRACT** Next-generation sequencing (NGS) enables rapid identification of common and rare drug-resistant genetic variations from tuberculosis (TB) patients' sputum samples and MTB isolates. However, whether this technology is effective for formalin-fixed and paraffin-embedded (FFPE) tissues remains unclear. An amplicon-based targeted NGS sequencing panel was developed to predict susceptibility to 9 antituberculosis drugs, including 3 first-line drugs, by directly detecting FFPE tissues. A total of 178 tissue samples from TB patients who underwent phenotypic drug susceptibility test were retrospectively tested from January 2017 to October 2019 in the Department of Pathology, Beijing Chest Hospital, China. Phenotypic drug susceptibility test results were used as the reference standard. We identified 22 high-quality mutations from 178 FFPE tissue samples, including 15 high+moderate+minimal confidence-level mutations associated with drug resistance (*rpoB* D435V, S450F/L; *KatG* S315T; *inhA-fabG* promoter c-15t; *embB* G406S, M306V; *rpsL* K43R, K88R, *rrs* a1401g, a514c; *gyrA* D94G/Y/A, A90V), 6 mutations not associated with resistance (*rpoB* D435Y, H445S, L430P, L452P; *embB* G406A/D), and one mutation site *embB* M306I defined as indeterminate. Compared to the phenotypic method, sensitivities (95% CI) for rifampicin, isoniazid, and ethambutol were 96% (79.65–99.90%), 93.55% (78.58–99.21%), and 71.43% (35.24–92.44%), respectively; while for second-line drugs, it varied from 23.53% (9.05–47.77%) for capreomycin to 86.84% (72.20–94.72%) for streptomycin. Specificities for all drugs were satisfactory (>94.51%). Therefore, important pathological FFPE tissue samples, despite partially degraded DNA, can be used as essential specimens for molecular diagnosis of drug resistant TB by amplicon-based targeted NGS technology.

**IMPORTANCE** Amplicon-based targeted NGS technology focuses on a set of gene mutations of known or suspected associations with drug susceptibility in *Mycobacterium tuberculosis* (MTB). This method offers many benefits, such as low sequencing cost, easy customization, high throughput, shorter testing time and not culture dependent. Formalin-fixed and paraffin-embedded (FFPE) tissues are important pathological specimen in diagnosing tuberculous disease because they are noninfectious and provide excellent preservation of tissue morphology with low storage cost. However, the performance of amplicon-based targeted NGS method on FFPE samples has not been reported yet. Therefore, we evaluated the performance of this method using FFPE samples collected from January 2017 to October 2019 in the Department of Pathology, Beijing Chest Hospital, China. We demonstrate that the amplicon-based targeted NGS method performs excellent on FFPE samples, and it can be applied to pathological diagnosis of drug resistant tuberculosis.

**KEYWORDS** amplicon-based targeted NGS, tuberculosis, drug resistant mutations, FFPE tissue samples

Address correspondence to Nanying Che, cheny0448@163.com.

The authors declare no conflict of interest.

**D**rug-resistant tuberculosis (DR-TB) is a devastating threat worldwide. The global incidence of MDR-TB is 3.4% in new cases and 18% in previously treated cases, while approximately 80% DR-TB patients cannot receive an appropriate drug regimen due to the lack of phenotypic drug susceptibility testing (DST) information (1). Culture-based DST is the gold standard to diagnose DR-TB for effective antituberculosis therapy. However, this method is time-consuming and requires stringent level of biosafety instruments (2).

The DR phenotype in *Mycobacterium tuberculosis* (MTB) is mainly determined by chromosomal mutations in several genes (3). For example, rifampicin resistance, caused by mutations in *rpoB* gene encoding the beta subunit of RNA polymerase, are the most common gene mutations in MTB. The Xpert MTB/RIF assay (Cepheid, USA) can detect both MTB and mutations in *rpoB* gene directly from sputum using PCR (PCR) technology. Yet, as this assay can only detect rifampicin-resistant mutations, Xpert MTB/XDR (Cepheid, USA) has been further updated to detect mutations associated with resistant to isoniazid, ethambutol, fluoroquinolones, and second-line injectable drugs (4). However, the target genes in Xpert MTB/XDR are still limited to only several genes and promoter regions. In addition, Xpert MTB/XDR cannot report the mutation types.

Whole-genome sequencing (WGS) of clinical MTB isolates allows for more accurate identification of all chromosomal mutations from a single test. This method has been widely adopted to analyze DR-TB across several countries or regions with great performance for first and second-line drugs (5–7). Nevertheless, WGS is only applicable for high-quality genomic DNA from MTB isolates, but not clinical samples. Additionally, although the culture of pulmonary TB has a higher sensitivity, the total sensitivity of MTB culture is only about 30% (1, 8) due to the much lower culture sensitivity of extrapulmonary TB, which largely limits the clinical application of WGS.

Alternatively, targeted next-generation sequencing (NGS) also enables rapid identification of common and rare genetic variations. It can focus on a select set of genes or gene regions of known or suspected associations with a specific pathogen (e.g., MTB) or a specific phenotype (e.g., drug resistance). It can be used on both MTB isolates and clinical specimens.

Formalin-fixed and paraffin-embedded (FFPE) sample is an important pathological specimen in diagnosing TB, especially for sputum-negative and extra-pulmonary TB. However, FFPE samples have some disadvantages compared to MTB isolates, such as additional steps for deparaffinization and the extracted DNA is partially degraded. Previous studies have shown that the MTB-specific gene fragment, *IS6110*, and the *rpoB* gene could be detected in FFPE samples using Xpert MTB/RIF method (9–10). Also, several mutation sites associated with resistance of rifampicin, isoniazid, ethambutol, and streptomycin, could be identified by MeltPro MTB assay in our previous study (11). Performances in molecular diagnosis of DR-TB are excellent with FFPE samples, which provides a significant clue for us to identify more mutation sites associated with multidrug resistance in MTB with this pathological specimen.

In this study, an amplicon-based targeted NGS panel was developed to detect gene mutations associated with drug resistance using FFPE samples. The diagnostic value of the amplicon-based targeted NGS technology developed for FFPE tissue sample was evaluated in diagnosing DR-TB.

## RESULTS

**Characteristics of participants.** Totally, 178 MTB patients (73 male/105 female, median age 48.9, range 30–67) were included in this study (Table 1), 55 of whom (55/178, 30.9%) were DR-TB patients. Phenotypic DST results for 178 TB patients were provided in Table S1.

Among the 55 DR-TB patients, they can be classified as 25 for rifampicin, 31 for isoniazid, 8 for ethambutol, 38 for streptomycin, 11 for kanamycin, 6 for amikacin, 17 for capreomycin, 12 for levofloxacin and 5 for moxifloxacin, as shown in Table 2.

**TABLE 1** Clinical characteristics of the participants

| Characteristic | Total (*n* = 178) | Drug-resistant phenotypic | Drug-sensitive phenotypic |
|---|---|---|---|
| Age, median (range) | 49 (12–84) | 39 (12–81) | 53 (17–84) |
| Sex, No. (%) | | | |
| Male | 105 (59.0) | 34 (19.1) | 71 (39.9) |
| Female | 73 (41.0) | 21 (11.8) | 52 (29.2) |
| Organ types, No. (%) | | | |
| Bone and joint | 96 (53.9) | 22 (12.4) | 74 (41.6) |
| Lung | 73 (41.0) | 30 (16.9) | 43 (24.2) |
| Pleura | 5 (2.8) | 1 (0.6) | 4 (2.2) |
| Lymph node | 4 (2.2) | 2 (1.1) | 2 (1.1) |

Additionally, the numbers of MDR (Multidrug Resistant) TB and XDR (Extensively Drug-Resistant) TB were 10 and 11, respectively.

**Analysis of mutation sites detected by amplicon-based targeted NGS.** The amplicon-based targeted NGS panel included 59 mutations in 11 gene regions. Practically, we detected 7 targeted gene regions (*rpoB*, *KatG*, *inhA-fabG* promoter, *embB*, *rpsL*, *rrs*, *gyrA*), in which 22 mutations were identified but 29 mutations were not. Of note, 8 mutations in *inhA*, *eis* promoter, *tlyA* or *gyrB* genes were not detected. Mutation sites detected by the amplicon-based targeted NGS for 178 TB patients were provided in Table S1. The most common mutation sites causing the resistance of rifampicin, isoniazid, ethambutol, amikacin, streptomycin, kanamycin, capreomycin, levofloxacin and moxifloxacin, respectively, were *rpoB* S450L (rifampicin), *KatG* S315T (isoniazid), *embB* M306V (ethambutol), *rrs* a1401g (amikacin), *rpsL* K43R (streptomycin), *rrs* a1401g (kanamycin), *rrs* a1401g (capreomycin), *gyrA* D94G (levofloxacin) and *gyrA* D94G (moxifloxacin).

According to the values and *P* values of LR and OR (Table S2), confidence level for each mutation site was graded based on the phenotypic DST results (Table 3). Eleven mutation sites were classified as high-confidence level, including *rpoB* D435V, S450F, S450L; *KatG* S315T; *embB* G406S; *rpsL* K43R, K88R; *rrs* a514c, a1401g; *gyrA* D94G, D94Y. Three mutation sites were classified as moderate-confidence level, including *embB* M306V; *gyrA* A90V, D94A. One mutation site, *inhA-fabG* promoter c-15t, was classified as minimal-confidence. It was worth noting that *rrs* a514c was highly associated with streptomycin resistance but displayed no association with kanamycin resistance. Besides, *gyrA* A90V was moderately associated with levofloxacin resistance, but not with moxifloxacin resistance. Six mutation sites, including *rpoB* D435Y, H445S, L430P, L452P and *embB* G406A, G406D, were not associated with any drug resistance, while drug association of *embB* M306I was indeterminate. Generally, mutations with high, moderate, and minimal confidence level were classified as DR-TB.

**TABLE 2** Sensitivity and specificity of the targeted NGS based drug resistance prediction

| Drug | No of phenotypic DST | | Sensitivity (%, 95% CI) | Specificity (%, 95% CI) | PPV[a] (%, 95% CI) | NPV[b] (%, 95% CI) |
|---|---|---|---|---|---|---|
| | Resistant | Susceptible | | | | |
| Rifampicin | 25 | 153 | 96.00 (79.65–99.90) | 100.00 (-[c]) | 100.00 (-) | 99.32 (95.89–99.99) |
| Isoniazid | 31 | 147 | 93.55 (78.58–99.21) | 94.56 (89.47–97.38) | 78.37 (62.56–88.86) | 98.58 (94.65–99.94) |
| Ethambutol | 8 | 170 | 71.43 (35.24–92.44) | 94.51 (89.76–97.22) | 35.71 (16.18–61.40) | 98.73 (95.18–99.95) |
| Streptomycin | 38 | 140 | 86.84 (72.20–94.72) | 95.71 (90.77–98.22) | 84.61 (69.89–93.14) | 96.40 (91.63–98.68) |
| Kanamycin | 11 | 167 | 36.36 (14.98–64.81) | 100.00 (-) | 100.00 (-) | 95.91 (91.64–98.16) |
| Amikacin | 6 | 172 | 66.67 (29.57–90.75) | 100.00 (-) | 100.00 (-) | 98.83 (95.56–99.95) |
| Capreomycin | 17 | 161 | 23.53 (9.05–47.77) | 99.37 (96.17–99.99) | 80.00 (35.96–97.97) | 92.39 (87.32–95.61) |
| Levofloxacin | 12 | 166 | 83.33 (54.00–96.5) | 94.58 (89.88–97.26) | 52.63 (31.70–72.67) | 98.74 (93.57–99.25) |
| Moxifloxacin | 5 | 173 | 80.00 (35.96–97.97) | 94.61 (89.94–97.28) | 30.77 (12.35–57.96) | 99.37 (96.17–99.0.99) |

[a]Positive predictive value.
[b]Negative predictive value.
[c]-, Indicates not available.

**TABLE 3** List of mutation sites identified by targeted NGS panel and their categories associated with phenotypic drug resistance

| Drug | Gene | High-confidence mutation | Moderate-confidence mutation | Minimal-confidence mutation | No association with resistance |
|---|---|---|---|---|---|
| Rifampicin | *rpoB* | D435V, S450F, S450L | -[a] | - | D435Y, H445S, L430P, L452P |
| Isoniazid | *KatG* | S315T | - | - | - |
| | *inhA-fabG* promoter | - | - | c-15t | - |
| | *inhA* | - | - | - | - |
| Ethambutol | *embB* | G406S | M306V | - | G406A, G406D |
| Streptomycin | *rpsL* | K43R, K88R | - | - | - |
| | *rrs* | a514c | - | - | - |
| Kanamycin | *rrs* | a1401g | - | - | a514c |
| | *eis* promoter | - | - | - | - |
| Amikacin | *rrs* | a1401g | - | - | a514c |
| Capreomycin | *rrs* | a1401g | - | - | - |
| | *tlyA* | - | - | - | - |
| Levofloxacin | *gyrA* | D94G, D94Y | A90V, D94A | - | - |
| | *gyrB* | - | - | - | - |
| Moxifloxacin | *gyrA* | D94G, D94Y | D94A | - | A90V |
| | *gyrB* | - | - | - | - |

[a]-, Indicates not available.

**Diagnostic performance of amplicon-based targeted NGS sequencing.** Diagnostic accuracy of the amplicon-based targeted NGS method was validated using the phenotypic DST as the reference (Table 2). The sensitivities for detecting rifampicin and isoniazid resistance were 96% (79.65–99.90%) and 93.55% (78.58–99.21%), with specificities of 100% and 94.56% (89.47–97.38%), respectively. The sensitivity of ethambutol, another first-line drug, was 71.43% (35.24–92.44%), with a specificity of 94.51% (89.76–97.22%). For the second-line drugs, the sensitivities varied from 86.84% (72.20–94.72%) for streptomycin to 23.53% (9.05–47.77%) for capreomycin, but their specificities were all above 94.50%.

Positive predictive values (PPV) of drugs involved in this study varied greatly (Table 2), but excellent performances were obtained in predicting drug resistance for rifampicin (100%) and isoniazid (78.37% [62.56–88.86%]). PPV for kanamycin, amikacin, streptomycin, capreomycin, levofloxacin, ethambutol and moxifloxacin were 100%, 100%, 84.61% (69.89–93.14%), 80% (35.96–97.97%), 52.63% (31.70–72.67%), 35.71% (16.18–61.40%) and 30.77% (12.35–57.96%), respectively. Negative predictive values (NPV) of all drugs were above 90%, indicating the panel's solid diagnostic capacity with low false positivity for detecting drug resistance.

## DISCUSSION

Gene mutations have been proved sufficient to reveal phenotypic drug resistance in MTB isolates. PCR-based method cannot rapidly get information associated with drug resistant in a high-throughput screening. With the rapid development of NGS in recent years, it has been recommended by WHO for the detection of mutations associated with drug resistance in MTB complex (12). In this study, the performance of amplicon-based targeted NGS sequencing panel was evaluated with pathological FFPE samples.

FFPE samples are important specimens for pathological diagnosis of TB because they are noninfectious, easy handling and long-term storage with low cost. However, the integrity of genome DNA in FFPE samples is compromised due to degradation and formalin will decrease the efficiency of PCR. The development of commercial reagents for DNA extraction from FFPE samples has unlocked the molecular diagnostic potential of this resource (13). PCR products between 100 and 300 bp can be generated from FFPE samples (14). Multiplex PCR procedure combined with minisequencing for high-

throughput single nucleotide polymorphism (SNP) has been reported excellent with 25-year-old FFPE tissues (15). As for TB disease, lesions are usually the reservoir for MTB survival, which makes the pathology archived FFPE samples from TB patients the ideal resource for molecular diagnosis of mutations related to drug resistance.

Compared with the performance of WGS using MTB isolates in previous study (7), our results manifest a high degree of accuracy of amplicon-based targeted NGS method for predicting susceptibility and resistance of TB patients to anti-TB drugs using FFPE samples. Sensitivities of rifampicin (96.00%) and isoniazid (93.55%) by the amplicon-based targeted NGS using FFPE samples were similar to that of WGS using MTB isolates (rifampicin, 97.5%; isoniazid 97.1%), while the sensitivity of ethambutol (71.43%) in this study was lower than that reported 94.6%. As for the second-line TB drugs, sensitivity of amplicon-based targeted NGS method was also comparable with that of WGS method using MTB isolate. The specificities of the amplicon-based targeted NGS panel using FFPE samples and those of WGS using MTB isolates were both satisfactory. These results demonstrate that amplicon-based targeted NGS is a promising method in molecular pathological diagnosis for drug resistant TB disease. However, one limitation of this study is the small sample size compared to the study carried out by Miotto and his colleagues (5), which will introduce some deviations.

Although the amplicon-based targeted NGS method exhibit excellent performances in predicting mutations related to drug resistance in MTB using isolates (16) and sputum samples (17), we cannot demonstrate that their panel is also applicable for FFPE samples. Our amplicon-based targeted NGS panel was specifically designed for FFPE samples considering the possible nontargeted bacterial genome. In clinic work, for sputum-negative pulmonary TB and extra-pulmonary TB patients, FFPE samples may be critical for diagnose. Our amplicon-based targeted NGS panel could further improve the positive results of drug resistant TB patients using FFPE samples.

WGS can verify a large number of DR-TB mutation sites associated with drug susceptibility, which is comparable with the phenotypic DST results (7). However, the application of WGS in detecting drug resistance in MTB requires genomic DNA from MTB isolates. What's more, WGS consumes a lot of time to process the huge sequencing data and requires professional bioinformatic skill to interpret the mutations. Amplicon-based targeted NGS focuses on genes and mutations of known with drug resistance, which offers many benefits, such as low sequencing cost, easy customization, high throughput, shorter testing time and not culture-dependent.

In conclusion, amplicon-based targeted NGS for FFPE samples is a rapid and accurate method to improve the ability for diagnosing drug resistant TB disease.

## MATERIALS AND METHODS

**Participants.** Between January, 2017, and October, 2019, 178 consecutive FFPE samples from 178 tuberculosis patients were tested retrospectively at Beijing Chest Hospital, Beijing, China. All the patients were culture positive, and DR-TB patients were further confirmed by culture-based DST assay. Amplicon-based targeted NGS results of drug susceptibility test were all available for patients included. Four sample types, bone and joint, lung, pleura and lymph node, were included in this study (Table 1).

**Amplicon-based targeted NGS panel.** A defined set of target mutation sites in 11 genes of MTB associated with 9 antituberculosis drugs were selected, including rifampicin, isoniazid, ethambutol, streptomycin, kanamycin, amikacin, capreomycin, levofloxacin, and moxifloxacin (Table S3). Strain H37Rv (GenBank accession no. NC_000962.3) was used as the reference genome to align sequenced amplicons. Several genes related to drug resistant in MTB have homologous genes in many other bacteria (3). Therefore, nontargeted bacteria in FFPE samples, regarded as background bacteria, can reduce the specificity of MTB amplification. *Sphingomonas*, *Enterococcus*, *Fusobacterium*, *Brevundimonas*, and *Streptococcus*, which have been widely reported in FFPE tissues (18–20), and those detected in our 8 FFPE samples were regarded as nontargeted bacteria in this study (Table S4). Position of each predefined mutation sites associated with drug resistance in the MTB H37Rv reference genome as well as genomes of nontargeted bacteria were submitted to the Ion AmpliSeq.Designer website for multiplex primer design (https://www.ampliseq.com/browse.action, Thermo Fisher Scientific Inc.). Finally, an amplicon-based targeted NGS panel of 39 primer pairs was generated to simultaneously detect multiple mutations related to drug resistance in MTB using FFPE samples. All oligonucleotides were synthesized in Thermo Fisher Scientific Inc. (Waltham, USA).

**Phenotypic drug susceptibility test (pDST).** Susceptibility of MTB strains to first- and second-line antituberculosis drugs were performed in the clinical laboratory of Beijing Chest Hospital using a

commercial 96-well plate containing lyophilized antibiotics according to manufacturer's instructions (Encode Medical Engineering Co, Zhuhai, China). Susceptibility was determined on the basis of critical concentration cutoffs (in parentheses) for the following drugs: rifampicin (1 mg/L), isoniazid (0.2 mg/L), ethambutol (2.5 mg/L), streptomycin (1 mg/L), kanamycin (2.5 mg/L), amikacin (1 mg/L), capreomycin (2.5 mg/L), levofloxacin (2 mg/L), moxifloxacin (0.5 mg/L).

**Experimental Procedures.** 10 4 $\mu$m FFPE slides were used to extract DNA using the FFPE DNA kit (Taipu Biosciences Co., Ltd., Beijing, China). Libraries were constructed using the AmpliSeq Library kit 2.0 (catalog no. 4471269; Life Technologies) with the customized primer pairs designed above. Sequencing was performed on the Ion Proton Sequencer (Thermo Fisher Scientific, Waltham, MA, USA). Called variants were accepted if sequencing depth was >100 and the allele frequency (AF) was >95%.

**Interpreting the association of mutations with phenotypic drug resistance.** Drug resistance associated mutations were graded according to a published standardized procedure (5). According to the $P$ values and values of both likelihood ratio (LR) and odds ratio (OR), mutations were classified into 3 levels of confidence for predicting resistance, including high, moderate, and minimal. No association was considered when the values <1 and $P$ values <0.05, while the indeterminate level was considered when the $P$ values ≥0.05. Values and $P$ values of LR and OR for high-quality mutation sites are provided in Table S2.

**Statistical analysis.** The sensitivity, specificity, positive predictive value (PPV) and negative predictive value (NPV) were evaluated with 95% confidence intervals using the SPSS Statistics 21.0 (SPSS) software. By comparison with phenotypic data, likelihood and odds ratio for each mutation sites were calculated in VassarStats (http://vassarstats.net/index.html).

**Ethical approval.** The study was approved by the Ethical and Institutional Review Boards for Human Investigation of the Beijing Chest Hospital (Number: 2020-keyan-linshen14).

**Data availability.** These original sequence data have been submitted to GenBank databases with accession number PRJNA771241.

## SUPPLEMENTAL MATERIAL

Supplemental material is available online only.
**SUPPLEMENTAL FILE 1**, XLSX file, 0.01 MB.
**SUPPLEMENTAL FILE 2**, XLSX file, 0.01 MB.
**SUPPLEMENTAL FILE 3**, XLSX file, 0.03 MB.
**SUPPLEMENTAL FILE 4**, PDF file, 0.2 MB.

## ACKNOWLEDGMENTS

This work was supported by grants from the National Natural Science Foundation of China (Grant Numbers: 82072381, 31900098), Beijing Municipal Science and Technology Project (Grant Numbers: Z181100001918027, Z191100006619079), and Tongzhou High-level Technique Talents Program (Grant Number: YHLD2018006). Beijing Municipal Administration of Hospitals Incubating Program (Grant Number: PX2020066).

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
