## [Reviewer comments · Microbiology Spectrum]

Microbiology Spectrum

Application of amplicon-based targeted NGS technology for diagnosis of drug-resistant tuberculosis using FFPE specimens

Nanying Che, Song Jing, Du Weili, Liu Zichen, Che Jialu, and Li Kun

Corresponding Author(s): Nanying Che, Department of Pathology, Beijing Key Laboratory for Drug Resistant Tuberculosis Research, Beijing Chest Hospital, Capital Medical University, Beijing Tuberculosis and Thoracic Tumor Research Institute, Beijing 101149, China

Review Timeline:

Submission Date:	September 1, 2021
Editorial Decision:	October 1, 2021
Revision Received:	December 8, 2021
Editorial Decision:	January 11, 2022
Revision Received:	January 13, 2022
Accepted:	January 18, 2022

Editor: William Lainhart

Reviewer(s): The reviewers have opted to remain anonymous.

Transaction Report:

DOI: <https://doi.org/10.1128/spectrum.01358-21>

October 1, 2021

Dr. Nanying Che

Department of Pathology, Beijing Key Laboratory for Drug Resistant Tuberculosis Research, Beijing Chest Hospital, Capital Medical University, Beijing Tuberculosis and Thoracic Tumor Research Institute, Beijing 101149, China
Beijing
China

Re: Spectrum01358-21 (Application of amplicon sequencing technology for diagnosis of drug-resistant tuberculosis using FFPE specimens)

Dear Dr. Nanying Che:

Thank you for submitting your manuscript to Microbiology Spectrum. When submitting the revised version of your paper, please provide (1) point-by-point responses to the issues raised by the reviewers as file type "Response to Reviewers," not in your cover letter, and (2) a PDF file that indicates the changes from the original submission (by highlighting or underlining the changes) as file type "Marked Up Manuscript - For Review Only". Please use this link to submit your revised manuscript - we strongly recommend that you submit your paper within the next 60 days or reach out to me. Detailed information on submitting your revised paper are below.

Link Not Available

Sincerely,

William Lainhart

Journals Department
Reviewer comments:

Reviewer #1 (Comments for the Author):

Jing Song, Weili Du, Nanying Che et. al. evaluated the use of NGS amplicon sequencing for detection of drug resistance mutations tb from FFPE specimens. I believe there is a clinical need for this type of assays specially in the absence of a Mtb isolate (i.e. culture were not performed or isolate did not grow). However, I feel the authors should provide more information on the NGS assay development as well as phenotypic testing (what method, which concentration(s) of drugs, liquid media and/or colonies) to better understand the correlation between both methods.

Please find additional comments below:

- 1) Line 68 "culturing Mycobacterium tuberculosis (MTB) is time-consuming..." The authors could emphasize that culture is still the gold standard
- 2) Line 74 "this assay targets only a few mutation sites in rpoB gene..." I would not say only a few mutations as it targets the RRDR region (where most rifampicin mutations are located).
- 3) Line 81 "The positive ratio of culture is very low". I don't understand what the authors meant. Please clarify.
- 4) Line 91. "require heavy experimental pretreatment". Please clarify.
- 5) Line 120-121 background bacterial taxa were considered to reduce the specificity of MTB amplification. Do the authors mean

- that non TB organisms genes could be amplified and sequenced along or instead those of MTB? Please rephrase and clarify.
- 6) Line 121-122 "Average relative abundance of Mycobacterium in FFPE tissue samples was 1.7% identified by 16S rRNA sequencing". How was this done experimentally? I would expect great variability in MTB abundance depending on sample type, smear positive vs negative, and patient clinical features.
 - 7) Line 122-123. "Organisms with relative abundance above 1.7% might interfere the amplification of gene fragment and were regarded as bacterial background in this panel" How did the authors reach this conclusion? is there experimental data to support it?
 - 8) Lines 135-137 "The copy number of IS6110 in elution buffer above 20/μl were required for DNA library construction. DNA would be pre-concentrated if the IS6110 copy number < 20/μl". How did the author measure IS6110? is there experimental data to support this cut off?
 - 9) Line 142-143. "the number of each amplicon > 100". Do the authors mean number of NGS reads for each amplicon? Please clarify.
 - 10) Line 151. "likelihood ratio and odds ratio" The reference Miotto et al 2017 seems to group both LR and OR together (in their Table 2). Please explain how each were calculated and include this information in the manuscript.
 - 11) Line 173-174 "Sixty-four of which (64/178, 35.95%) were DR-TB patients resistant to at least one antibiotic drug". It's not clear to me what the criteria for a sample to be "DR-TB". Is monoresistance = DR-TB? Also the authors indicate 64 DR-TB patients, but Table 1 shows different numbers: 57 (34 male + 23 female) and 59 (sum samples per organ type).
 - 12) Line 179-180 "Additionally, XDR (resistant to any fluoroquinolones or at least one of the second-line injectable drugs)" I would say (MDR + resistant to any fluoroquinolones or at least one of the second-line injectable drugs)
 - 13) Line 189-190. "6 mutations not associated with resistance (rpoB D435Y, H445S, L430P, L452P; embB G406A/D), and 2 contradictory mutations (rrs a514c; gyrA A90V)" and Table 2. This is an example why it would be helpful to know how phenotypic drug testing was performed. D435Y, L430P, L452P were associated with low level rifampicin resistance (pubmed.ncbi.nlm.nih.gov/32999007/) while G406A/D was reported as likely associated with ethambutol resistance (pubmed.ncbi.nlm.nih.gov/26033726/; SNPs sites i.e. <https://card.mcmaster.ca/ontology/39910>). Please explain the meaning of "contradictory" mutations.
 - 14) Line 239-240. "the major mutation site of rpoB, S450L and less frequently mutated, D435V". I would emphasize that S450L is among the most common rifampicin resistance mutations globally.
 - 15) Line 261-264. I believe the discussion should be expanded comparing this work to other publications on NGS and drug resistance done on FFPE (if available) and to NGS and drug resistance done directly on clinical samples (not FFPE) and on clinical isolates, with a summary of advantages and disadvantages of each approach.
 - 16) Line 267-269. The authors allude to background taxa here (and earlier in the text). However, they do not provide further explanations nor supportive data.
 - 17) Table 2. Please explain why c-15t appears as minimal-confidence mutation in this manuscript but as moderate-confidence mutation by Miotto et al.
 - 18) Fig 1. Figure 1 Percentage of mutation sites identified for each anti tuberculosis drug. Font is very small. Also lower frequency mutations appear as 0 (so can't tell their actual frequency).

Minor comments:

- 1) Line 93. "reduces noises". I'd say noise (instead of noises)
- 2) Line 108. "were collected retrospectively". I would say were tested retrospectively.

Reviewer #2 (Comments for the Author):

The manuscript by Song and Du et al. employ a targeted amplicon sequencing platform of common drug resistance determining regions in formalin-fixed and paraffin-embedded (FFPE) samples with *M. tuberculosis*. FFPE samples are important pathological specimen for the diagnosis of TB and in this case, drug resistant *Mtb*. Overall, the manuscript is straightforward. I do have a few comments and suggestions:

- Line 71: Xpert MTB/RIF assay does only detect polymorphisms in rpoB. However, Xpert MTB/XDR extends the drug targets to isoniazid, fluoroquinolones, second-line injectable drug (amikacin, kanamycin, capreomycin) and ethionamide in a single test. While this does not detract from the findings of this study, they should be contextualized.
- Lines 183: This section in the Results describe the mutations in FFPE samples. Reporting mutations is appropriate however the study is not well suited for asserting whether mutations are associated with phenotypic resistance. More so, no statistical testing is provided.
- Line 183, are the rpoB mutants listed from single strains? Do these strains have additional mutations in rpoB? Or are all strains harboring these mutations susceptible based on Rif DST? It would be helpful to have a list of all drug resistant strains (n=64), their phenotypic susceptibility and associated polymorphism data.
- The Discussion is much too long with lengthy detailing of mutations and phenotypic confidence, distracting from the main message that is amplicon seq is a robust strategy for detecting polymorphisms in drug resistance determining regions in FFPE samples.
- Overall the manuscript could be greatly shortened to focus on the reliability of detecting mutations in known regions of resistance alongside phenotypic data.

Staff Comments:

Preparing Revision Guidelines

Please return the manuscript within 60 days; if you cannot complete the modification within this time period, please contact me. If you do not wish to modify the manuscript and prefer to submit it to another journal, please notify me of your decision immediately so that the manuscript may be formally withdrawn from consideration by Microbiology Spectrum.

Manuscript ID: Spectrum01358-21

Title: Application of amplicon sequencing technology for diagnosis of drug-resistant tuberculosis using FFPE specimens

Dear Dr. Lainhart,

Thanks very much for the valuable comments on our manuscript. We have taken all the reviewers' comments into account and revised our manuscript. Point-by-point response to reviewer's comments was given below.

Thank you very much for paying attention to our manuscript.

Sincerely yours,

Nanying Che

Response to Reviewer1

1. I feel the authors should provide more information on the NGS assay development as well as phenotypic testing (what method, which concentration(s) of drugs, liquid media and/or colonies) to better understand the correlation between both methods.

Re: We provide more details, such as the identification of the non-targeted bacteria, on the NGS assay development, please see Line 231-248. Susceptibility of MTB strains to first- and second-line antituberculosis drugs were performed in clinical laboratory of Beijing Chest Hospital using a commercial 96-well plates containing lyophilized antibiotics according to manufacturer's instructions (Encode Medical Engineering Co, Zhuhai, China). Please see Line 251-258.

1. Line 68 "culturing *Mycobacterium tuberculosis* (MTB) is time-consuming..." The authors could emphasize that culture is still the gold standard.

Re: As recommended, we revised the sentences as "*Culture-based DST is the gold standard to diagnose DR-TB for effective anti-TB therapy. However, this method is time-consuming and requires stringent level of biosafety instruments.*" at Line 76-79.

2. Line 74 "this assay targets only a few mutation sites in *rpoB* gene..." I would not say only a few mutations as it targets the RRDR region (where most rifampicin mutations are located).

Re: We revised the sentence according to the recommendation. We want to emphasize that the main shortcomings of the Xpert is detecting only mutations related to rifampicin resistance. Also, we added contexts on Xpert MTB/XDR (Cepheid, USA). Please see Line 85-88 "Yet, as this assay can only detect rifampicin-resistant mutations, Xpert MTB/XDR (Cepheid, USA) has been further updated to detect resistance-associated mutations of isoniazid, ethionamide, fluoroquinolones, and second-line injectable drugs."

3. Line 81 "The positive ratio of culture is very low". I don't understand what the authors meant. Please clarify.

Re: As recommended, this sentence has been revised as "*the sensitivity of MTB culture is only about 30%*" at Line 96-97.

4. Line 91. "require heavy experimental pretreatment". Please clarify.

Re: This has been revised as "However, FFPE samples have some disadvantages compared to MTB isolates, such as additional steps for deparaffinization and the extracted DNA is partially degraded", please see Line 105-106.

6. Line 120-121 background bacterial taxa were considered to reduce the specificity of MTB amplification. Do the authors mean that non TB organisms genes could be amplified and sequenced along or instead those of MTB? Please rephrase and clarify.

7. Line 121-122 "Average relative abundance of *Mycobacterium* in FFPE tissue samples was 1.7% identified by 16S rRNA sequencing". How was this done experimentally? I would expect great variability in MTB abundance depending on sample type, smear positive vs negative, and patient clinical features.

8. Line 122-123. "Organisms with relative abundance above 1.7% might interfere the amplification of gene fragment and were regarded as bacterial background in this panel" How did the authors reach this conclusion? is there experimental data to support it?

17. Line 267-269. The authors allude to background taxa here (and earlier in the text). However, they do not provide further explanations nor supportive data.

Response for 6, 7, 8, 17:

1) Several genes related to drug resistance in MTB have homologous genes in many other bacteria. Non TB organisms in FFPE samples, regarded as background bacteria, can reduce the specificity of MTB amplification.

2): *Sphingomonas*, *Enterococcus*, *Fusobacterium*, *Brevundimonas* and *Streptococcus* have been reported in FFPE tissues.

3): Bacteria described in 2) were also detected in our 8 FFPE tissue samples through 16S rRNA sequencing method. Relative abundance of *Mycobacterium* varied greatly from 0% to 7.65%. The average relative abundance of *Mycobacterium* was 1.7% across the 8 FFPE tissues. Bacteria with relative abundance above 1.7% together accounted for > 85.49% of the total 16S rRNA sequences in FFPE samples, suggesting that most non TB bacteria in FFPE samples were considered.

Therefore, based on the bacteria taxa reported in published literature and our laboratory experiments, bacterial background were determined in Table S4.

9. Lines 135-137 "The copy number of IS6110 in elution buffer above 20/μl were required for DNA library construction. DNA would be pre-concentrated if the IS6110 copy number < 20/μl". How did the author measure IS6110? is there experimental data to support this cut off?

Re: The copy number of *IS6110* in elution buffer was measured based on standard curves according to manufacturer's instructions (Sansure Biotech, Beijing, China). The cut off value (20 copies/μl) was identified based on the protocol for DNA library construction (AmpliSeq™ Library Kit 2.0; catalog no. 4471269; Life Technologies). In this study, amplicon-based targeted NGS results of drug susceptibility test were all available for the TB patients.

10. Line 142-143. "the number of each amplicon > 100". Do the authors mean number of NGS reads for each amplicon? Please clarify.

Re: This sentence means "the number of NGS reads for each amplicon > 100". This has been revised as "Called variants were accepted if sequencing depth was >100 and the allele frequency (AF) was >95%" at Line 265-266.

11. Line 151. "likelihood ratio and odds ratio" The reference Miotto et al 2017 seems to group both LR and OR together (in their Table 2). Please explain how each were calculated and include this information in the manuscript.

Re: Our data was calculated according to the method raised by Miotto et al., 2017. We calculate values and p values of LR and OR in an online website VassarStats (<http://vassarstats.net/index.html>). This website was stressed at Line 282. Details of the statistical analysis for all mutations are provided in Table S2.

In this manuscript, we adopted the proposed confidence levels for grading mutations with phenotypic resistance in the reference Miotto et al., 2017. When both P-values of LR

and OR are <0.05 , both values of LR and OR are >10 , the mutation is defined as high confidence for association with resistance; both values are 5-10, the mutation is defined as moderate confidence for association with resistance; both values are 1-5, the mutation is defined as minimal confidence for association with resistance. No association was considered when both the values <1 and P values <0.05 , while the indeterminate level was considered when the P values ≥ 0.05 .

12. Line 173-174 "Sixty-four of which (64/178, 35.95%) were DR-TB patients resistant to at least one antibiotic drug". It's not clear to me what the criteria for a sample to be "DR-TB". Is monoresistance = DR-TB?

Re: DR-TB means patients resisted at least one antibiotic drug according to the phenotypic DST results.

Also the authors indicate 64 DR-TB patients, but Table 1 shows different numbers: 57 (34 male + 23 female) and 59 (sum samples per organ type).

Re: Based on the phenotypic DST results, 55 DR-TB patients were identified. The numbers in Table 1 was revised.

13. Line 179-180 "Additionally, XDR (resistant to any fluoroquinolones or at least one of the second-line injectable drugs)" I would say (MDR + resistant to any fluoroquinolones or at least one of the second-line injectable drugs)

Re: This sentence has been revised at Line 127-128.

14. Line 189-190. "6 mutations not associated with resistance (rpoB D435Y, H445S, L430P, L452P; embB G406A/D), and 2 contradictory mutations (rrs a514c; gyrA A90V)" and Table 2. This is an example why it would be helpful to know how phenotypic drug testing was performed. D435Y, L430P, L452P were associated with low level rifampicin

resistance (pubmed.ncbi.nlm.nih.gov/32999007/) while G406A/D was reported as likely associated with ethambutol resistance (pubmed.ncbi.nlm.nih.gov/26033726/; SNPs sites i.e. <https://card.mcmaster.ca/ontology/39910>).

Please explain the meaning of "contradictory" mutations.

Re: Phenotypic DST for each MTB isolates were performed using a drug susceptibility test kit for *Mycobacteria* (Encode, Zhuhai, China), and detailed information was provided in Line 251-258.

With DST results, drug resistant associations of each detected mutation sites were determined based on the p-values and values of LR and OR raised by Miotto et al 2017. Mutations sites *rpoB* D435Y, *rpoB* L430P, *rpoB* L452P and *embB* G406A/D were classified as no association with drug resistance.

In the study (pubmed.ncbi.nlm.nih.gov/32999007/), association with low level rifampicin resistance of *rpoB* D435Y, L430P and L452P are determined when the MTB isolates exhibit elevated rifampicin MICs compared to fully susceptible strains but remain phenotypically susceptible by mycobacterial growth indicator tube (MGIT) testing and have been associated with poor patient outcomes.

Ethambutol resistant associated mutation sites *embB* G406A/D was determined by likelihood ratio test in (pubmed.ncbi.nlm.nih.gov/26033726/) and (SNPs sites i.e. <https://card.mcmaster.ca/ontology/39910>).

Conclusively, there are two aspects that can explain this difference. 1) Different evaluation criteria might directly lead to the discrepant drug associations of a mutation site. 2) Our sample size is small, which will also introduce some deviations.

As for "contradictory", we want to stress the mutations that show statistical drug association with one drug but show no drug association with another drug. In this study, *rrs* a514c was associated with streptomycin resistance in a high level confidence, but was not associated with kanamycin resistance. *gyrA* A90V was associated with levofloxacin resistance in a moderate level confidence, but was not associated with moxifloxacin resistance. These two mutations were revised as confidence-level mutations associated with drug resistance.

15. Line 239-240. "the major mutation site of *rpoB*, S450L and less frequently mutated, D435V". I would emphasize that S450L is among the most common rifampicin resistance mutations globally.

Re: As recommended, this sentence has been revised at Line 136-139.

16. Line 261-264. I believe the discussion should be expanded comparing this work to other publications on NGS and drug resistance done on FFPE (if available) and to NGS and drug resistance done directly on clinical samples (not FFPE) and on clinical isolates, with a summary of advantages and disadvantages of each approach.

Re: We revised the discussion according to the recommendation. Please see Discussion.

18. Table 2. Please explain why c-15t appears as minimal-confidence mutation in this manuscript but as moderate-confidence mutation by Miotto et al.

Re: This can be explained from two aspects: (1) This study is carried out with Chinese population, and the degrees of the drug resistant association for this mutation site might vary in different populations. (2) Our sample size is small compared to the study carried out by Miotto et al, which will introduce some deviations.

19. Fig 1. Figure1 Percentage of mutation sites identified for each anti tuberculosis drug. Font is very small. Also lower frequency mutations appear as 0 (so can't tell their actual frequency).

Re: As recommended, descriptions related to Figure 1 was revised and Figure 1 was deleted in the manuscript.

Minor comments:

1) Line 93. "reduces noises". I'd say noise (instead of noises)

Re: As recommended, this phrase has been revised.

2) Line 108. "were collected retrospectively". I would say were tested retrospectively.

Re: As recommended, the word "collected" has been revised as "tested" at Line 224.

Reviewer #2 (Comments for the Author):

The manuscript by Song and Du et al. employ a targeted amplicon sequencing platform of common drug resistance determining regions in formalin-fixed and paraffin-embedded (FFPE) samples with *M. tuberculosis*. FFPE samples are important pathological specimen for the diagnosis of TB and in this case, drug resistant *Mtb*. Overall, the manuscript is straightforward. I do have a few comments and suggestions:

1. - Line 71: Xpert MTB/RIF assay does only detect polymorphisms in *rpoB*. However, Xpert MTB/XDR extends the drug targets to isoniazid, fluoroquinolones, second-line injectable drug (amikacin, kanamycin, capreomycin) and ethionamide in a single test. While this does not detract from the findings of this study, they should be contextualized.

Re: As recommended, besides Xpert MTB/RIF assay, characteristics of Xpert MTB/XDR assay were also contextualized in the manuscript at Line 85-88.

2. - Lines 183: This section in the Results describe the mutations in FFPE samples. Reporting mutations is appropriate however the study is not well suited for asserting whether mutations are associated with phenotypic resistance. More so, no statistical testing is provided.

Re: We graded mutations associated with phenotypic resistance according to a standardized method proposed by Miotto et al., 2017. Values and p values for LR (Likelihood ratio) and OR (Odds Ratio) for all high-qualified mutation sites were provided in Table S2.

3. - Line 183, are the *rpoB* mutants listed from single strains? Do these strains have additional mutations in *rpoB*? Or are all strains harboring these mutations susceptible based on Rif DST? It would be helpful to have a list of all drug resistant strains (n=64), their phenotypic susceptibility and associated polymorphism data.

Re: Thanks for your recommendation.

We tested mutation sites in MTB genome using FFPE samples, not MTB strains. The *rpoB* mutants listed are a summary of all the 178 samples.

Phenotypic DST results and mutation sites detected by amplicon-based targeted NGS panel for 178 TB patients were provided in Table S1.

4. - The Discussion is much too long with lengthy detailing of mutations and phenotypic confidence, distracting from the main message that is amplicon seq is a robust strategy for detecting polymorphisms in drug resistance determining regions in FFPE samples.
5. - Overall the manuscript could be greatly shortened to focus on the reliability of detecting mutations in known regions of resistance alongside phenotypic data.

Response for 4, 5:

As recommended, associations between mutations and phenotypic confidence were shortened. Reliability of the amplicon-based targeted NGS panel for detecting mutations with FFPE samples was elaborated in discussion.

January 11, 2022

Dr. Nanying Che

Department of Pathology, Beijing Key Laboratory for Drug Resistant Tuberculosis Research, Beijing Chest Hospital, Capital Medical University, Beijing Tuberculosis and Thoracic Tumor Research Institute, Beijing 101149, China

9# Beiguandajie

Beijing

China

Re: Spectrum01358-21R1 (Application of amplicon-based targeted NGS technology for diagnosis of drug-resistant tuberculosis using FFPE specimens)

Dear Dr. Nanying Che:

Thank you for submitting your manuscript to Microbiology Spectrum. As you will see your paper is very close to acceptance. Please modify the manuscript along the lines I have recommended. As these revisions are quite minor, I expect that you should be able to turn in the revised paper in less than 30 days, if not sooner. If your manuscript was reviewed, you will find the reviewers' comments below.

When submitting the revised version of your paper, please provide (1) point-by-point responses to the issues I raised in your cover letter, and (2) a PDF file that indicates the changes from the original submission (by highlighting or underlining the changes) as file type "Marked Up Manuscript - For Review Only". Please use this link to submit your revised manuscript. Detailed instructions on submitting your revised paper are below.

Link Not Available

Sincerely,

William Lainhart

Reviewer comments:

Reviewer #1 (Comments for the Author):

This is a revised version of a manuscript by Jing Song, et. al. on amplicon-based NGS for diagnosis of drug-resistant tuberculosis using FFPE specimens. I believe the authors have addressed most of the reviewers' concerns. However, I still have some comments as shown below (mostly English and Grammar usage and one comment regarding limitations of the study)

Line numbers are based on companion_ms file

Line 56 "including 15 confidence-level mutations" a word seems to be missing between 15 and confidence
Line 73 "free cultivation". I think the author meant that it's free from having to culture the organism. I would say "not culture-dependent"

Line 133 "non-infectious, excellent". I would say non-infectious, and provide excellent

Line 180 "the sensitivity of MTB culture is only about 30% (1, 8)" I would emphasize that this low % of culture sensitivity applies to extrapulmonary TB. Culture of pulmonary TB has higher sensitivity.

Line 284-287. I suggest to include the drug following each mutation, for better clarity. For example: "The most common antibiotic resistance conferring mutations were rpoB S450L (rifampicin), katG S315T (isoniazid), etc"

Line 398 "drug resistant in a high-throughput" A word seems to be missing after high-throughput

Line 484 "integrity of genome DNA in FFPE samples is seriously degraded" I'd say the integrity of genome DNA in FFPE samples is compromised due to degradation

Line 507 "we cannot demonstrate that their panel also applicable for FFPE samples". A verb is missing between panel and also.
Line 513 "Whatever, " I would not use the word Whatever
Line 515 "This is to say". I would start the sentence with "Our amplicon-based targeted.. " (remove This is to say)
Line 525-526 "In conclusion, amplicon-based targeted NGS for FFPE samples is a rapid and accurate method to improve the ability for diagnosing TB disease." This last sentence is very general; I would focus on drug resistance, which is the topic of this study.

The authors have included in the response to reviewers ""Our sample size is small compared to the study carried out by Miotto et al, which will introduce some deviations." This is useful information that would be good to include in the discussion (sample size as a limitation of this study)

Preparing Revision Guidelines

- point-by-point responses to the issues I raised in your cover letter
- Upload a compare copy of the manuscript (without figures) as a "Marked-Up Manuscript" file.
- Each figure must be uploaded as a separate file, and any multipanel figures must be assembled into one file.
- Manuscript: A .DOC version of the revised manuscript
- Figures: Editable, high-resolution, individual figure files are required at revision, TIFF or EPS files are preferred

Please return the manuscript within 60 days; if you cannot complete the modification within this time period, please contact me. If you do not wish to modify the manuscript and prefer to submit it to another journal, please notify me of your decision immediately so that the manuscript may be formally withdrawn from consideration by Microbiology Spectrum.

Manuscript ID: Spectrum01358-21

Title: Application of amplicon-based targeted NGS technology for diagnosis of drug-resistant tuberculosis using FFPE specimens

Dear Dr. Lainhart,

Thanks very much for the valuable comments on our manuscript. We have taken all the reviewers' comments into account and revised our manuscript. Point-by-point response to reviewer's comments was given below.

Thank you very much for paying attention to our manuscript.

Sincerely yours,

Nanying Che

Reviewer comments:

Reviewer #1 (Comments for the Author):

This is a revised version of a manuscript by Jing Song, et. al. on amplicon-based NGS for diagnosis of drug-resistant tuberculosis using FFPE specimens. I believe the authors have addressed most of the reviewers' concerns. However, I still have some comments as shown below (mostly English and Grammar usage and one comment regarding limitations of the study)

Line numbers are based on companion_ms file

Line 56 "including 15 confidence-level mutations" a word seems to be missing between 15

and confidence

Re: As recommended, this phrase was revised as “including 15 high+moderate+minimal confidence-level mutations” at Line 41-42.

Line 73 "free cultivation". I think the author meant that it's free from having to culture the organism. I would say "not culture-dependent"

Re: As recommended, we have revised “free cultivation” as “not culture-dependent” at Line 59.

Line 133 "non-infectious, excellent". I would say non-infectious, and provide excellent

Re: As recommended, this sentence has been revised as “Formalin-fixed and paraffin-embedded (FFPE) tissues are important pathological specimen in diagnosing tuberculous disease because they are non-infectious, and provide excellent preservation of tissue morphology with low storage cost.” at Line 61-62.

Line 180 "the sensitivity of MTB culture is only about 30% (1, 8)" I would emphasize that this low % of culture sensitivity applies to extrapulmonary TB. Culture of pulmonary TB has higher sensitivity.

Re: As recommended, this sentence has been revised as “although the culture of pulmonary TB has a higher sensitivity, the total sensitivity of MTB culture is only about 30% (1, 8) due to the much lower culture sensitivity of extrapulmonary TB.” at Line 97-99.

Line 284-287. I suggest to include the drug following each mutation, for better clarity. For example: "The most common antibiotic resistance conferring mutations were rpoB S450L (rifampicin), katG S315T (isoniazid), etc"

Re: Drug has been added in the parenthesis following each mutation according to the recommendation at Line 140-143.

Line 398 "drug resistant in a high-throughput" A word seems to be missing after high-throughput

Re: As recommended, this sentence been revised as "PCR-based method cannot rapidly get information associated with drug resistant in a high-throughput screening." at Line 179.

Line 484 "integrity of genome DNA in FFPE samples is seriously degraded" I'd say the integrity of genome DNA in FFPE samples is compromised due to degradation

Re: As recommended, this sentence has been revised as "the integrity of genome DNA in FFPE samples is compromised due to degradation" at Line 186.

Line 507 "we cannot demonstrate that their panel also applicable for FFPE samples". A verb is missing between panel and also.

Re: As recommended, this has been revised as "we cannot demonstrate that their panel is also applicable for FFPE samples" at Line 211-212.

Line 513 "Whatever, " I would not use the word Whatever

Re: As recommended, the word "Whatever" has been removed. Please see Line 213-214.

Line 515 "This is to say". I would start the sentence with "Our amplicon-based targeted.. "
(remove This is to say)

Re: As recommended, the phrase "This is to say" has been removed. Please see Line

215.

Line 525-526 "In conclusion, amplicon-based targeted NGS for FFPE samples is a rapid and accurate method to improve the ability for diagnosing TB disease." This last sentence is very general; I would focus on drug resistance, which is the topic of this study.

Re: According to the recommendation, the last sentence has been revised as "In conclusion, amplicon-based targeted NGS for FFPE samples is a rapid and accurate method to improve the ability for diagnosing drug resistant TB disease." Please see Line 226.

The authors have included in the response to reviewers ""Our sample size is small compared to the study carried out by Miotto et al, which will introduce some deviations." This is useful information that would be good to include in the discussion (sample size as a limitation of this study)

Re: Thanks for your recommendation, the sentence "However, one limitation of this study is the small sample size compared to the study carried out by Miotto and his colleagues (5), which will introduce some deviations." has been added in the discussion at Line 207-208.

January 18, 2022

Dr. Nanying Che
Department of Pathology, Beijing Key Laboratory for Drug Resistant Tuberculosis Research, Beijing Chest Hospital, Capital Medical University, Beijing Tuberculosis and Thoracic Tumor Research Institute, Beijing 101149, China
9# Beiguandajie
Beijing
China

Re: Spectrum01358-21R2 (Application of amplicon-based targeted NGS technology for diagnosis of drug-resistant tuberculosis using FFPE specimens)

Dear Dr. Nanying Che:

Your manuscript has been accepted, and I am forwarding it to the ASM Journals Department for publication. You will be notified when your proofs are ready to be viewed.

Sincerely,

William Lainhart
Editor, Microbiology Spectrum

Journals Department
Supplemental Material: Accept
Supplemental Material: Accept
Supplemental Material: Accept
Supplemental Material: Accept